# A study on measuring the $^{222}$Rn in the Buriganga River and tap water of the megacity Dhaka

**M. S. Alam**[1], **M. M. Mahfuz Siraz**[2], **Jubair A. M.**[1], **S. C. Das**[3], **D. A. Bradley**[4,5], **Mayeen Uddin Khandaker**[5,6], **Shinji Tokonami**[7], **Afroza Shelley**[1], **Selina Yeasmin**[2]*

**1** Department of Nuclear Engineering, University of Dhaka, Dhaka, Bangladesh, **2** Health Physics Division, Atomic Energy Centre, Dhaka, Bangladesh, **3** Institute of Nuclear Minerals, Bangladesh Atomic Energy Commission, Savar, Dhaka, Bangladesh, **4** Centre for Nuclear and Radiation Physics, Department of Physics, University of Surrey, Guildford, Surrey, United Kingdom, **5** Centre for Applied Physics and Radiation Technologies, School of Engineering and Technology, Sunway University, Bandar Sunway, Selangor, Malaysia, **6** Department of General Educational Development, Faculty of Science and Information Technology, Daffodil International University, Dhaka, Bangladesh, **7** Institute of Radiation Emergency Medicine, Hirosaki University, Hirosaki, Japan

* selinayeasmin@yahoo.com

**Data Availability Statement:** All the data are available within the paper.

**Funding:** The authors received no specific funding for this work.

## Abstract

Radon ($^{222}$Rn), an inert gas, is considered a silent killer due to its carcinogenic characteristics. Dhaka city is situated on the banks of the Buriganga River, which is regarded as the lifeline of Dhaka city because it serves as a significant source of the city's water supply for domestic and industrial purposes. Thirty water samples (10 tap water from Dhaka city and 20 surface samples from the Buriganga River) were collected and analyzed using a RAD H$_2$O accessory for $^{222}$Rn concentration. The average $^{222}$Rn concentration in tap and river water was 1.54 ± 0.38 Bq/L and 0.68 ± 0.29 Bq/L, respectively. All the values were found below the maximum contamination limit (MCL) of 11.1 Bq/L set by the USEPA, the WHO-recommended safe limit of 100 Bq/L, and the UNSCEAR suggested range of 4–40 Bq/L. The mean values of the total annual effective doses due to inhalation and ingestion were calculated to be 9.77 μSv/y and 4.29 μSv/y for tap water and river water, respectively. Although all these values were well below the permissible limit of 100 μSv/y proposed by WHO, they cannot be neglected because of the hazardous nature of $^{222}$Rn, especially considering their entry to the human body via inhalation and ingestion pathways. The obtained data may serve as a reference for future $^{222}$Rn-related works.

## 1. Introduction

Humans are continuously exposed to natural radiation, primarily from terrestrial and extra-terrestrial sources [1]. Among the existing sources of ionizing radiation in the environment, $^{222}$Rn alone is the major contributor (more than 50%) of the total radiation dose to humans [2]. $^{222}$Rn is the only gaseous element in the $^{238}$U decay series and possesses no color, odor, or taste. This ($^{222}$Rn) short-lived ($T_{1/2}$ = 3.82 days) radioactive nucleus is formed due to the alpha

**Competing interests:** The authors have declared that no competing interests exist.

decay of $^{226}$Ra. Among the three naturally occurring radioisotopes, $^{222}$Rn is the most abundant in nature as Thoron ($^{220}$Rn) and Actinon ($^{219}$Rn) have relatively very short half-lives of 55s and 3.2s, respectively.

$^{222}$Rn is present naturally in the earth's strata. Its abundance in the earth's crust fluctuates with the variation of geology and lithology of the area. Due to its high mobility, $^{222}$Rn gas can swiftly travel from soil and rocks to water and air. Albeit, the concentration of $^{222}$Rn in water depends on the temperature, lithology, geology, rainfall, and earthquake activities [3]. $^{222}$Rn is highly volatile, easily dissolved, and escapes from the water. A relatively higher concentration of $^{222}$Rn is found in groundwater than in surface water due to the aeration process [3]. Because of its gaseous nature, $^{222}$Rn is used as a tectonic tracer [4] to determine the tectonic fault lines and predict earthquakes.

$^{222}$Rn is considered a hazardous gas due to its potential to affect human cells and tissues biologically. Ingestion through the gastrointestinal tract and inhalation via the respiratory tract are the two major pathways of entering $^{222}$Rn into the human body. Both paths are potentially risky, affecting the lung and the gastrointestinal system. In the case of inhalation, the short-lived metallic progeny of $^{222}$Rn (mostly $^{218}$Po and $^{214}$Po) are deposited in the lungs and damage the cells and the tissues of the respiratory system via high-energy alpha emission. That is why it is one of the main contributors to escalating lung cancer risks. The IARC (The International Agency for Research on Cancer), a part of WHO, classified $^{222}$Rn as a group 1 carcinogen [5, 6].

Water is vital for all life; human beings use water regularly for various purposes, including bathing, drinking, etc. However, water consumption is the primary cause of $^{222}$Rn exposure through the ingestion pathway, whereas the emanation of $^{222}$Rn from water causes exposure through the inhalation of air. As $^{222}$Rn is loosely soluble in water, it can easily emanate from water to air [7]. For that reason, $^{222}$Rn activity measurement in water is necessary to protect people from radiological hazards. Many international organizations propose a safe limit on $^{222}$Rn concentration in water, and almost all developed countries have their national guidelines for radiation safety. The World Health Organization recommended a safe limit of 100 Bq/L for $^{222}$Rn in the water [8], whereas the USEPA suggested the maximum contamination level (MCL) of 11.1 Bq/L [9]. The USEPA also proposes an alternative maximum contamination level (AMCL) of 148 Bq/L [9]. To apprehend the health hazard of $^{222}$Rn, measurement of the annual effective dose due to $^{222}$Rn ingestion and inhalation is essential. The WHO recommends that the total annual effective dose due to $^{222}$Rn in water should be < 100 μSv [10].

Numerous studies have been performed worldwide to measure the $^{222}$Rn level in various water resources such as tap water, river water, deep well water, bore well water, bottled water, etc. [1, 3, 7, 11, 12]. Several advanced countries have a national reference limit of radon in water and indoor air to ensure radiological safety for public health. Bangladesh has no such reference level for $^{222}$Rn in water. Millions of people living in the Dhaka megacity solely rely on tap water for their daily household purposes, such as washing, bathing, drinking, cooking, etc. The Buriganga river serves as one of the busiest major transportation routes/hubs, as well as many businesses and trade centers that are situated on the bank of this river. This indicates a greater possibility of $^{222}$Rn exposure to the general populace. So, it is necessary to measure the $^{222}$Rn level in the tap water and the Buriganga river water to find out if it is within the safe limit or not, which eventually will help to ensure the radiological safety of public health.

The purpose of this study is to (a) measure $^{222}$Rn concentration in the chemically and biologically polluted Buriganga river water and the tap water of the megacity Dhaka, b) calculate the associated radiological hazards, c) to contribute to the setting up of a factual baseline data which will assist the authority to structure a national reference level of $^{222}$Rn water.

## 2. Methodology

### 2.1 Study area

Dhaka, the capital city of Bangladesh, as well as one of the most densely populated megacities in the world, is the prime focus of this study. Dhaka is located between latitudes 23˚42' and 23˚54'N and longitudes 90˚20' and 90˚28'E. The geographical area of this city is 306.38 square kilometers, where more than 20 million people [13]. Several rivers, like Buriganga, Balu, Tongi Khal, and Turag, surround the city from the south, east, west, and north [14], respectively. However, the Buriganga river has a major share, and it forms the southern and western boundaries of Dhaka city. The length of this river flowing through Dhaka is 11 km, the depth is 10m, and the width is 400m. The latitude and the longitude of this river are 23˚ 37' 59.99" N, 90˚ 25' 59.99" E [15]. Because of the large-scale industrial activities on the bank of the Buriganga river, it has become the worst polluted river in the country.

### 2.2 Geology of Dhaka city and its periphery region

Dhaka, the megacity, is placed at the southern end of the Madhupur tract, 1.5–10 m (average 6 m) above the adjoining floodplains [16, 17]. The area is characterized by Quaternary alluvial sequences of the Madhupur Tract, known as Pleistocene terrace deposits that surround Holocene deposits of the peripheral rivers [18–20]. The geological map of the study area is illustrated in Fig 1(b), showing different geological units present in this area. The Pleistocene terrace deposits of varying thickness (an average of 10 m thick in Dhaka) are subdivided by Upper and Lower Madhupur Clay deposits. The Upper Madhupur Clay deposits are

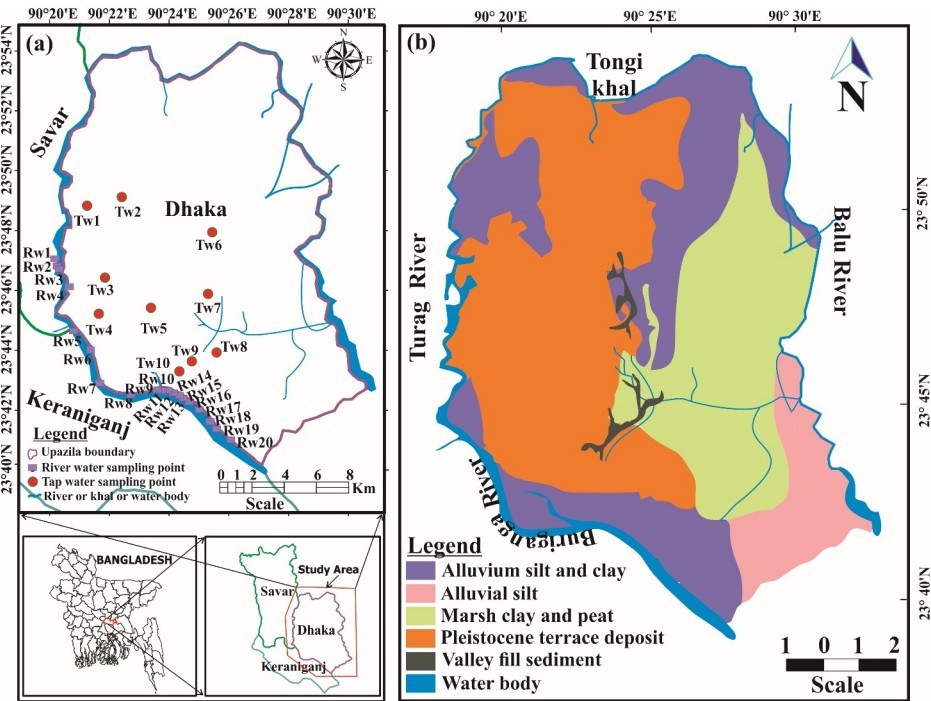

**Fig 1.** (a) The study area map showing the sampling location of river water (RW) and tap water (TW). The map has been produced using ArcGIS 10.4.1 software. The sources of basemaps of administrative boundaries and inland water bodies: Esri, GADM, Garmin, DigitalGlobe, GeoEye, GEBCO, USGS, NOAA, National Geographic, EPA, Geonames. org, the GIS User Community and other contributors. (b) The geology of the study area (the sources of basemaps are similar to (a) and the geological units modified after [16, 28, 29]).

**Table 1. Lithostratigraphy of the study area (after [17, 21]).**

| Age | Formation | Lithology | Aquifer characteristics | Thickness (m) |
|---|---|---|---|---|
| Holocene | Basabo (Alluvium) | Silt and clay with discontinuous sand | Linked to surface drainage | 2–5 |
| Pleistocene | Madhupur clay | Silty clay and fine sand | Aquitard | 12–15 |
| Pliocene | DupiTila | Sand with a discontinuous silty clay layer | Aquifer (upper) | 2500 |
| | | | Aquitard (middle) | |
| | | | Aquifer (lower) | |

characterized by reddish brown to pale yellow sticky clay and silty clay, containing ferruginous nodules and dark spots of manganese, compacted highly weathered and oxidized residual deposits. On the other hand, the Lower Madhupur Clay deposits primarily contain pale yellowish to yellowish brown sandy clay to clayey sand and silty sand with similar nodules and spots but less weathered and oxidized than the upper [16, 17]. The Holocene deposits are further subdivided into alluvial floodplain deposits comprising natural levee deposits, bar deposits, point bar deposits, back swamp deposits, floodplain deposits, and valley fill deposits. Floodplain deposits mainly comprise grey to dark grey color sticky clay to clayey silt, with discontinuous sand, oxidized root, rootlets, and organic matter. Whereas the valley fill deposits consist of dark grey to yellowish to olive brown color silty clay, clay, marshy clay, and peat [21]. A sequence of fine to coarse-grained micaceous quartzofeldspathic sands containing Dupi Tila Formation of Pliocene age, hydro geologically known as the Dupi Tila aquifers, the primary aquifer of Dhaka city, underlies the Madhupur Clay and is not exposed anywhere in the city [17, 19, 20, 22]. A gravel bed lies at the bottom of the Dupi Tila Formation, which grades upward from coarse-grained sands to medium-grained sands to fine-grained sands at the top. The Dupi Tila Formation is divided by a discontinuous clay layer into two aquifers: an upper fine-grained aquifer (approximately 40–50 m thick) and a lower coarse-grained aquifer (approximately 80 m thick) [17]. A summary of the Pliocene to Recent lithological and aquifer characteristics of the study area has been given in Table 1. The geochemical study of the groundwater of the Dupi Tila aquifer shows that the Ca/Mg-HCO3 type and weathering of aluminusilicates control the distribution of major ions in the aquifers [23]. The Dupi Tila and Madhupur Formations are isolated by extensive incision of the land surface during the late Quaternary, and forming a number of faults at their boundaries which affect the aquifer river system and the groundwater flow of this area [19–21, 24, 25]. It is assumed that due to the elevation of the river bed with the top of the Dupi Tila sands has through connection between the and the rivers surrounding Dhaka (i.e. Buriganga, Balu and Turag River) and the aquifer is possible along certain reaches [17, 26, 27].

## 2.3 Sampling

Thirty water samples, including 20 river water and 10 tap water (Fig 1a), were collected in November 2021 using a 500 mL plastic bottle prior to the winter season. The river water samples were collected from the highly polluted Buriganga river by following the stratified sampling technique approved by IAEA [30]. The majority of the samples were collected from heavily populated riverbank areas such as Sadarghat, Showari Ghat, Mitford Ghat, Gabtoli, etc. The bottle was fully submerged directly into the water during the river water collection to prevent air bubbles in the bottle. The tap waters were collected from different localities of the megacity Dhaka using a systematic grid sampling technique approved by the IAEA [30]. Before sample collection, the tap was opened for several minutes, and the water was allowed to flow. Afterward, the bottle was filled and sealed tightly. Prevention of aeration during sampling

was the prime concern to avoid the escape of dissolved [222]Rn in the water. Each of the samples was labeled with a unique sample ID (RW for river water and TW for tap water), and the GPS of the collection points and the collection time were recorded. These water samples were taken immediately to the Laboratory of the Health Physics Division in the Atomic Energy Centre Dhaka of Bangladesh Atomic Energy Commission.

## 2.4 Experimental procedure

The [222]Rn activity concentration in collected water samples was measured using RAD7, a portable electric [222]Rn detector with RAD-H$_2$O accessories (manufactured by Durridge Co. Ltd). The RAD H$_2$O is an accessory of the RAD7 detector that allows measuring radon in water at concentrations above the minimum detectable activity (MDA). The MDA concentration of this instrument is 0.004 Bq/L [5, 31]. A schematic diagram of the experimental setup is illustrated in Fig 2. The inner cell of the RAD7 is a hemisphere coated with an electrical conductor where the energy of emitted alpha particles from [222]Rn and its progeny are converted into electrical signals. Before analyzing the samples, the RAD7 detector needs to be [222]Rn free and dry. Dry air was purged for 10 minutes, lowering the humidity below 10%. The collected water samples were transferred into a 250 mL glass vial and connected with the RAD7. The [222]Rn emanation occurred by aerating the water via a glass frit in a closed-loop system. An internal air circulating pump recirculates the air through the closed-loop system to extract the [222]Rn from the water until the equilibrium is reached. The wat-250 process was selected to measure [222]Rn in water, where the extraction efficiency was 94%. The equilibrium state is reached within 5 minutes, and after this, no more [222]Rn can be extracted from the water. The air is circulated by the pump aerating the water and supplying the [222]Rn to the RAD7 detector. This process runs for 30 minutes in four cycles to measure the [222]Rn in the samples. The RAD7 summarizes the average and corrected [222]Rn concentration measurements obtained from each sample for four cycles at the end of the run in a printout. For a cycle when no counts were

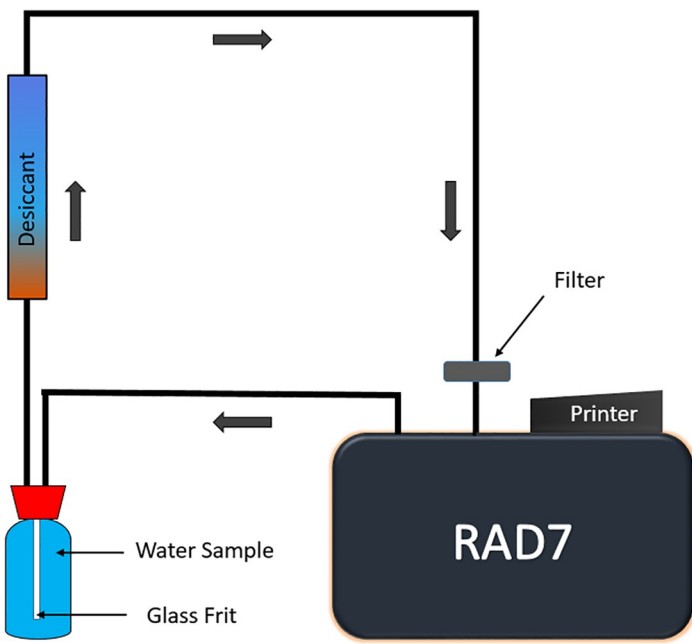

**Fig 2. Schematic diagram of RAD-H$_2$O detector closed loop aeration system, where the air and water volume remain constant and independent of the flow rate.**

collected, the RAD7 displays an uncertainty value based on a two-sigma, 95% confidence interval that is equivalent to ± 4 counts [32].

## 2.5 Dosimetry calculation

Internal $^{222}$Rn exposure comes primarily from $^{222}$Rn inhalation and ingestion, which is harmful to the respiratory organs. When water is collected and used, $^{222}$Rn is inhaled, and $^{222}$Rn is ingested when $^{222}$Rn-contaminated water is consumed. Therefore, by using Eqs (1) and (2), the annual effective dose due to $^{222}$Rn inhalation and ingestion is calculated from the experimentally measured values of the $^{222}$Rn concentration [1, 2, 5, 7]. The total annual effective dose is calculated by using Eq (3).

$$\sum D_{ig}(\mu Sv/y) = C_{RnW} \times C_W \times 365 \times EDC \times 10^{-3} \qquad (1)$$

$$\sum D_{in}(\mu Sv/y) = C_{RnW} \times R_{AW} \times F \times O \times DCF \qquad (2)$$

$$\text{Total Annual Effective Dose}(\mu Sv/y) = \sum D_{ig} + \sum D_{in} \qquad (3)$$

Where,

$\Sigma D_{ig}$ and $\Sigma D_{in}$ represents effective doses due to ingestion and inhalation, respectively

$C_{RnW}$ = $^{222}$Rn activity concentration in collected samples measured by RAD-H$_2$O detector (Bq/L)

$C_W$ (Daily water consumption) = 3 L/day [1, 10, 33]

EDC (Effective Dose Coefficient) = 3.5 nSv/Bq for $^{222}$Rn ingestion [2]

$10^{-3}$ is used for the conversion of nano-to-micro

$R_{AW}$ (ratio of $^{222}$Rn in the air to water) = $10^{-4}$ [2]

F (equilibrium factor between $^{222}$Rn and its progeny) = 0.4 [2]

O (mean indoor occupancy factor) = 7000 h/y [2]

DCF (dose conversion factor for $^{222}$Rn exposure) = 9 nSv(hBqm$^{-3}$)$^{-1}$ [2]

## 2.6 Ethics approval

This is an observational study. The Atomic Energy Centre Dhaka Research Ethics Committee has confirmed that no ethical approval is required.

# 3. Results and discussion

## 3.1 $^{222}$Rn in river water

As demonstrated in Table 2, the measured $^{222}$Rn concentration in the collected twenty river water samples from the highly polluted Buriganga river varied from 0.35 ± 0.18 to 1.16 ± 0.61 Bq/L with an average of 0.68 ± 0.29Bq/L. The maximum $^{222}$Rn concentration (1.16 ± 0.61 Bq/L) was found in the sample collected from the Forashgonj Kheyaghat area (RW17). There is a direct swage-drain line (from the Dolai Khal) near the Forashgonj Kheyaghat, which may contaminate the area with technologically enhanced naturally occurring radioactive materials (TENORMs), consequently may increase the $^{222}$Rn level in that location. The sample collected

**Table 2. Measured $^{222}$Rn concentration and calculated effective doses for the Buriganga river water.**

| Sample ID | Location Near | Latitude | Longitude | Mean Concentration (Bq/L) | Annual Effective Dose of Ingestion (μSv/y) | Annual Effective Dose of Inhalation (μSv/y) | Total Annual Effective Dose (μSv/y) |
|---|---|---|---|---|---|---|---|
| RW01 | Aminbazar Bridge | 23.7843125 | 90.3362020 | 0.63 ± 0.27 | 2.41 | 1.59 | 4.00 |
| RW02 | Gabtoli Balughat 2 | 23.7802673 | 90.3374692 | 0.66 ± 0.14 | 2.53 | 1.67 | 4.20 |
| RW03 | Gabtoli Balughat 1 | 23.7789416 | 90.3385024 | 0.70 ± 0.41 | 2.68 | 1.76 | 4.44 |
| RW04 | Azim Tower | 23.7687041 | 90.3442110 | 0.91 ± 0.49 | 3.49 | 2.29 | 5.78 |
| RW05 | Basila Bridge | 23.7442182 | 90.3466333 | 0.63 ± 0.14 | 2.41 | 1.59 | 4.00 |
| RW06 | Jhauchar Ghat | 23.7336485 | 90.3561759 | 0.56 ± 0.36 | 2.15 | 1.41 | 3.56 |
| RW07 | Gudara Ghat | 23.7145463 | 90.3612411 | 0.52 ± 0.31 | 1.99 | 1.32 | 3.31 |
| RW08 | Jadbar Bazar Ghat | 23.7085396 | 90.3786198 | 0.53 ± 0.24 | 2.03 | 1.32 | 3.35 |
| RW09 | Showari Ghat | 23.7113611 | 90.3947077 | 0.73 ± 0.18 | 2.80 | 1.85 | 4.65 |
| RW10 | Imamgonj Ghat | 23.7113133 | 90.3965892 | 0.49 ± 0.18 | 1.88 | 1.24 | 3.12 |
| RW11 | Mitford Ghat | 23.7109461 | 90.3994460 | 0.94 ± 0.50 | 3.60 | 2.38 | 5.98 |
| RW12 | Mitford Hospital Ghat | 23.7109609 | 90.3994378 | 0.46 ± 0.07 | 1.76 | 1.15 | 2.91 |
| RW13 | Babu Bazar Terminal | 23.7095952 | 90.4025324 | 0.52 ± 0.38 | 1.99 | 1.32 | 3.31 |
| RW14 | Badamtoli Ghat | 23.7082794 | 90.4051203 | 0.91 ± 0.33 | 3.49 | 2.29 | 5.78 |
| RW15 | Wais Ghat | 23.7066571 | 90.4078869 | 0.35 ± 0.18 | 1.34 | 0.88 | 2.22 |
| RW16 | Sadarghat | 23.7045832 | 90.4114031 | 0.42 ± 0.12 | 1.61 | 1.06 | 2.67 |
| RW17 | Forashgonj Kheyaghat | 23.7002854 | 90.4167060 | 1.16 ± 0.61 | 4.45 | 2.92 | 7.37 |
| RW18 | Dhaka Saw Mill | 23.6937433 | 90.4228607 | 0.81 ± 0.21 | 3.10 | 2.03 | 5.13 |
| RW19 | Postogola Bridge | 23.6898054 | 90.4263290 | 0.80 ± 0.21 | 3.07 | 2.03 | 5.10 |
| RW20 | Shyampur fire service | 23.6835213 | 90.4344459 | 0.77 ± 0.37 | 2.95 | 1.94 | 4.89 |
| **Average** | | | | **0.68 ± 0.29** | **2.59** | **1.70** | **4.29** |
| **Minimum** | | | | **0.35 ± 0.18** | **1.34** | **0.88** | **2.22** |
| **Maximum** | | | | **1.16 ± 0.61** | **4.45** | **2.92** | **7.37** |

from the Wais Ghat area (RW15) had the minimum $^{222}$Rn concentration (0.35 ± 0.18 Bq/L). The $^{222}$Rn level in these river water samples is relatively low as the aeration of surface water accelerates the emanation of $^{222}$Rn into the environment [5, 34]. No sample either contained a $^{222}$Rn concentration level more than the safe limit of 100 Bq/L recommended by the WHO or exceeded the maximum contamination limit (MCL) of 11.1 Bq/L set by USEPA [8, 9]. The obtained radon concentrations were also below the UNSCEAR suggested range of 4–40 Bq/L [35].

For each river water sample, the annual dose due to $^{222}$Rn inhalation and ingestion is listed in Table 2. The mean annual effective dose due to river water ingestion and inhalation were 2.59 μSv/y and 1.70 μSv/y, respectively. The total annual effective dose for river water ranged from 2.22 μSv/y to 7.38 μSv/y with an average of 4.29 μSv/y. All of these values were well below the maximum permissible limit of 100 μSv/y set by WHO [8].

In Table 3, the present study for river water is compared with the reported results world-wide. The $^{222}$Rn level was found very high in some river water, such as the $^{222}$Rn level (60 Bq/L) in the Rajouri of Pir Panjal, Kashmir was high due to the mountainous area where many

**Table 3. A worldwide comparative scenario of the $^{222}$Rn level in river water.**

| Country/ Region | Mean $^{222}$Rn Concentration (Bq/L) | Reference |
|---|---|---|
| Peninsular, Malaysia | 5.04 | [5] |
| Kwara,Nigeria | 15.97 | [34] |
| Ekiti, Nigeria | 42.22–88.22 | [3] |
| Edu, Nigeria | 19.14 ± 3.98 | [37] |
| Punjab, India | 3.37 ± 0.29 | [1] |
| Rajouri, Pir Panjal | 60 | [36] |
| Kirkuk, Iraq | 0.359 | [40] |
| Hemavathi River, India | 0.67 | [39] |
| Transylvania, Romania | 0.9–4.5 | [41] |
| Karnataka, India | 0.16–1.79 | [38] |
| Dhaka, Bangladesh | 0.68 ± 0.29 | Present work |

minerals were found in the soil of that region [36]. The study at Ekiti, Nigeria, claimed that the high $^{222}$Rn level (42–88 Bq/L) was found in river water due to the local geology covered with migmatite, porphyritic granite, granite gneiss, and undifferentiated schist [3]. In another study, the authors claimed the Gold and Bismuth mining site near the study area in Edu LGA, Kwara State, Nigeria, was the main reason for the high $^{222}$Rn level (19.14 ± 3.98 Bq/L) [37]. Nevertheless, the geological map of the Buriganga shows that there are no mountains or volcanic areas around this river. Neither any mining site nor the study area was covered with minerals. The Buriganga riverbed is mainly clay instead of rocks [17, 26, 27]. These were the significant reasons for the low $^{222}$Rn level in this river water. Additionally, the result of this study is consistent with the previous research carried out in different regions of the world, such as in Karnataka, India (0.16–1.79 Bq/L) [38], Hemavathi River India (0.67 Bq/L) [39], Kirkuk, Iraq (0.359Bq/L) [40].

### 3.2 $^{222}$Rn in tap water

As illustrated in Table 4, the $^{222}$Rn concentration in the ten tap water samples collected from Dhaka city varied from 0.56 ± 0.30 to 3.06 ± 0.60 Bq/L with an average of 1.54 ± 0.38 Bq/L.

**Table 4. Measured $^{222}$Rn concentration and calculated effective doses for the tap water.**

| Sample ID | Location Near | Latitude | Longitude | Mean Concentration (Bq/L) | Annual Effective Dose of Ingestion (μSv/y) | Annual Effective Dose of Inhalation (μSv/y) | Total Annual Effective Dose (μSv/y) |
|---|---|---|---|---|---|---|---|
| TW01 | Rupnagar | 23.8140410 | 90.3542840 | 2.45 ± 0.71 | 9.39 | 6.17 | 15.56 |
| TW02 | Mirpur 11 | 23.8187101 | 90.3736211 | 1.64 ± 0.32 | 6.29 | 4.13 | 10.42 |
| TW03 | Shyamoli | 23.7738063 | 90.3641953 | 0.91 ± 0.18 | 3.49 | 2.29 | 5.78 |
| TW04 | Mohammadpur | 23.7537414 | 90.3607778 | 3.06 ± 0.61 | 11.73 | 7.71 | 19.44 |
| TW05 | Farmgate | 23.7570560 | 90.3898441 | 1.19 ± 0.35 | 4.56 | 3.00 | 7.56 |
| TW06 | Baridhara | 23.7991200 | 90.4240744 | 2.59 ± 0.48 | 9.93 | 6.53 | 16.45 |
| TW07 | Rampura | 23.7648065 | 90.4217147 | 0.63 ± 0.08 | 2.41 | 1.59 | 4.00 |
| TW08 | Kamalapur | 23.7321814 | 90.4264804 | 1.61 ± 0.37 | 6.17 | 4.06 | 10.23 |
| TW09 | Gulistan | 23.7272202 | 90.4126653 | 0.74 ± 0.40 | 2.84 | 1.86 | 4.70 |
| TW10 | Bongshal | 23.7215800 | 90.4057534 | 0.56 ± 0.30 | 2.15 | 1.41 | 3.56 |
| Average | | | | 1.54 ± 0.38 | 5.89 | 3.88 | 9.77 |
| Minimum | | | | 0.56 ± 0.30 | 2.15 | 1.41 | 3.56 |
| Maximum | | | | 3.06 ± 0.61 | 11.73 | 7.71 | 19.44 |

The lowest $^{222}$Rn concentration (0.56 ± 0.30 Bq/L) was found in the Bongshal area (TW10). The sample collected from the Mohammadpur area (TW04) contained the highest $^{222}$Rn concentration (3.06 ± 0.61 Bq/L). A thorough investigation found that deep tube well water was stored in a tank and then supplied to the tap in the house from where the TW04 was collected. The water was stored in a closed tank that prevented air contact with water. For this reason, the $^{222}$Rn gas hardly emanates from the water, so the $^{222}$Rn level was higher than the others. However, all the samples contained lower $^{222}$Rn levels than both the maximum contamination limit (MCL) of 11.1 Bq/L set by USEPA and the safe limit of 100 Bq/L recommended by the WHO [11, 42–44].

The annual effective dose due to $^{222}$Rn inhalation and ingestion for each tap water sample is listed in Table 3. The maximum and the minimum values of annual effective dose due to tap water ingestion were 11.73 µSv/y and 2.15 µSv/y, with an average of 5.89 µSv/y. For inhalation, it ranged from 1.41 µSv/y to 7.71µSv/y with a mean of 3.87 µSv/y. The total annual effective dose for tap water ranged from 3.56 µSv/y and 19.44 µSv/y with an average of 9.77 µSv/y. All of these values were way below the maximum permissible limit of 100 µSv/y set by WHO [8].

Table 5 compares the present study for tap water with the reported literature worldwide. According to the previous literature, a high $^{222}$Rn level in tap water was found in some countries. A study in the Sabzevaran fault, Iran, found the $^{222}$Rn level in tap water higher (17.12 Bq/L) than in the MCL. The authors concluded that a high $^{222}$Rn level was due to volcanic, metamorphic, and sedimentary rocks surrounding the study area [45]. A study in Kenya found the $^{222}$Rn level (37 Bq/L) much higher than the MCL in tap water samples; due to the studied area being located near a volcanic region and the maximum tap water of the area was collected from underground water sources, the local geology was the primary reason for the abnormally higher $^{222}$Rn level [12].

The tap water of Dhaka city is collected from surface water treatment plants as well as extracted underground water by using different pumps [54], which are then supplied all over the city through a piping system. However, the geology of the present study area neither consisted of any volcanic, granitic, or metamorphic rock nor any volcanic region nearby. Therefore, these may be the leading causes of the lower $^{222}$Rn level in the tap water of Dhaka city.

**Table 5. A worldwide comparative scenario of the $^{222}$Rn level in tap water.**

| Country/ Region | Mean $^{222}$Rn Concentration (Bq/L) | Reference |
|---|---|---|
| Penang, Malaysia | 0.066 | [46] |
| Bitlis, Turkey | 0.59 to 66.00 | [42] |
| Xinjiang, China | 0.543 | [11] |
| Chiang Mai, Thailand | 0.18–1.13 | [7] |
| Sabzevaran fault, Iran | 17.12 | [45] |
| Zarand, Iran | 5.16 to 14.4 | [47] |
| Kabini River Basin, India | 8.5 | [48] |
| Sik, Malaysia | 0.0171 ± 0.0036 | [4] |
| Giresun University, Turkey | 0.98 to 27.28 | [49] |
| Rajasthan, India | 0.5 to 15 | [50] |
| Kedah, Malaysia | 7.0 ± 0.71 | [51] |
| Bihor, Romania | 6.9 | [52] |
| Nablus, Palestine | 1.0 | [43] |
| Bursa, Turkey | 0.91 to 12.58 | [53] |
| Kenya | 37 | [12] |
| Dhaka, Bangladesh | **1.53 ± 0.38** | Present work |

Moreover, the result of this study is consistent with many studies conducted in China [11], Thailand [7], Palestine [43], Malaysia [46], and India [48].

The present study shows that the [222]Rn level in river water is much lower than in tap water. River water is easily in contact with the open air, which accelerates the emanation of [222]Rn, while tap water has less contact with the air. Tap water is supplied in a closed piping system from the storage tank to the tap, so the aeration is negligible compared to surface water. Additionally, a portion of the tap water of Dhaka city is supplied from a groundwater source which was the primary reason for the high [222]Rn level in some samples like TW04.

## 3.3 Radiological risks based on geology of the study area

Radon emanates from soils, rocks, alluvial sediments, and/or aquifer matrices and enters the groundwater and air. Radon, the major contributor to natural background radiation exposure, and its progenies such as [218]Po, [214]Po, and [214]Bi release energetic alpha particles (high linear energy transfer) after inhalation and/or ingestion, causing lining in the stomach and lung cancer in the human body. Therefore, considering the health effect of radon, it is important to identify the areas with high radon concentration, their source, and relation with local geology to prevent the adverse effects on human being and the environment [55]. Though radon ([222]Rn) and thoron ([220]Rn) occurs naturally in most soils, sediments, and rocks as a radioactive decay product of [238]U of [232]Th respectively, the amount differs with localities and geological materials. Radon potential depends on the concentration of naturally occurring radionuclides such as [238]U or [226]Ra and [232]Th in the soils and types of bedrock present in the area [56]. Different geological factors such as lithology/rock type, porosity, permeability, compaction, emanation capacity of the ground, soil constituents, and tectonic features like faults, thrust, and joints, along with the geochemical and hydrogeological conditions of the area mainly control the source, distribution, transport and migration of radon in the soils, sediments, and rocks [57–59]. Certain rock types such as granites, metamorphosed granitic rocks, phosphate rocks with enriched uranium, coal deposits, black shale fractured/faulted rocks, and the subsequent soils resulting from these rocks are the most common sources of radon gas [56, 59, 60]. On the other hand, quartzose sandstone, non-organic shales, and siltstones are the least likely sources of radon [61], but under a favorable reducing environment, uranium mineralization may occur in alluvial-type sedimentary deposits which can then contain and emanate radon [59]. Based on the above facts and the geology of the study area, the radon potential and their associated health risks are evaluated in this study. Geologically, the study area mainly consists of Pleistocene terrace deposit (mixture of clay, silt and sand), alluvium silt, clay, mash clay, peat, valley fill deposits and bar sand (Fig 1b). A limited number of studies on distribution of NORMs in soils of Dhaka city and its surrounding areas and their radiological risks are available in the literature. Miah et al., 1998, studied on the distribution of radionuclides in soil samples in and around Dhaka city and found that the activity concentrations of [226]Ra, [232]Th and [40]K varied as 21–43 Bq/kg, 9–22 Bq/kg, and 165–750 Bq/kg, respectively, and except [40]K, the values of [226]Ra and [232]Th are below the world average [62]. On the other hand, the average background radiation dose level in and around Dhaka City shows 2.00 ± 0.47 mSv/y over a period of ten years, from 2006 to 2015, and demonstrates that no appreciable shift was seen even after the Fukusima Daiichi nuclear power plant disaster in Japan [63]. Therefore, due to the presence of low concentration of radioactive materials such as [226]Ra and [232]Th in the soils of Dhaka city and its periphery environs, mostly alluvial and clayey type sediments and/or soil, liberation of diffused radon in the atmosphere resulting a relatively lower concentration of radon in the associated river and tap water.

The Stochastic radiation model is based on the probabilistic nature of radiation-induced cancer and suggests that there is no threshold limit for radiation exposure below which the

risk of cancer becomes zero. This means that even a single atom of $^{222}$Rn in water can potentially cause hazard to the body by ionizing molecules and damaging cellular structures. Therefore, it is important to closely monitor the levels of $^{222}$Rn in water, as even a low concentration can pose a risk to human health. Despite all measured values of $^{222}$Rn in the tap and river water of Dhaka city show below the limit set by the USEPA and WHO, continuous monitoring is essential to ensure that the levels remain within the safe limits. The USEPA limit for $^{222}$Rn in drinking water is 11.1 Bq/L, while the WHO guideline value is 100 Bq/L. In this study, the measured levels of $^{222}$Rn in tap water and river water ranged from 0.56 ± 0.30 to 3.06 ± 0.61 Bq/L and from 0.35 ± 0.18 to 1.16 ± 0.61 Bq/L, respectively. The corresponding effective doses were found to be below the limit of 0.1 mSv/y recommended by the WHO [8]. Nevertheless, given the potential health risks associated with even little concentration of $^{222}$Rn in water, continuous monitoring of its concentration is essential to ensure safety of public health.

Many advanced countries have established national reference limits for radon in water and indoor air in order to ensure radiological safety and protect public health. However, Bangladesh currently lacks such a reference level for $^{222}$Rn in water, despite millions of people in the Dhaka megacity relying solely on tap water for daily household activities, including washing, bathing, drinking, and cooking. Given that the Buriganga River serves as a major transportation hub and facilitates too many businesses and trade centers, there is a greater likelihood of $^{222}$Rn exposure for the general population. In terms of concentration, it has been observed in this study that the tap water have a higher concentration of $^{222}$Rn than the river water. This is because, radon in river water can be easily diluted due to greater surface and interactions. However, this can vary depending on factors such as the local geology and the treatment processes employed for tap water. When it comes to dose, the risk of exposure to $^{222}$Rn from tap water is greater than from river water, since people are likely consume more tap water than river water. However, exposure to $^{222}$Rn in river water can still occur through activities such as swimming, fishing, etc. Overall, from an environmental and scientific viewpoint, it is important to monitor the concentration of $^{222}$Rn in both tap water and river water to ensure that exposure levels do not exceed the safe limits. This can help to protect public health and ensure that the water that use for daily activities is safe and free from harmful contaminants. Therefore, this study measures the $^{222}$Rn levels in both tap water and Buriganga river water to determine if they fall within the safe limits and ultimately ensure the radiological safety of the public. This study on $^{222}$Rn levels in Dhaka city's water may provide a valuable insight for future research on radiation exposure and human health.

## 4. Conclusion

In this study in Dhaka city, $^{222}$Rn level in river and tap water was measured using a RAD H$_2$O accessory to ensure public health safety from radiological hazards due to radon. The ranges of measured $^{222}$Rn concentrations in the river water (0.35 ± 0.18 to 1.16 ± 0.61 Bq/L) and the tap water (0.56 ± 0.30 to 3.06 ± 0.61 Bq/L) showed lower than the limit set by the WHO and the USEPA [44]. Also, the total annual effective doses were within the safe limit set by the WHO [8]. Considering the carcinogenic characteristics of $^{222}$Rn, frequent monitoring of $^{222}$Rn in various dwelling media is essential to ensure public health safety. Further extensive research should be carried out for $^{222}$Rn mapping of the country.

A few recommendations are proposed for future $^{222}$Rn-related works,

- Expansion of the study to other regions of Bangladesh is essential to obtain a better understanding on the distribution of $^{222}$Rn in different environmental media, including water, air, and soil.

- It is necessary to investigate the potential impact of local geological factors, such as soil type and groundwater composition, on the levels of $^{222}$Rn in water resources.

- Exploration of the relationship between $^{222}$Rn exposure and cancer incidence rates in the region is crucial to gain a better understanding on the implications of the public health.

- A risk assessment study should be conducted to evaluate the potential health risks associated with the chronic exposure to low levels of $^{222}$Rn in water resources in Dhaka and other regions of Bangladesh.

- A long-term monitoring program needs to be developed and implemented to track the changes of $^{222}$Rn levels over time and ensure that the safety of public health remain effective.

## Author Contributions

**Conceptualization:** M. S. Alam, M. M. Mahfuz Siraz, Mayeen Uddin Khandaker.

**Data curation:** M. S. Alam, M. M. Mahfuz Siraz.

**Formal analysis:** M. S. Alam, M. M. Mahfuz Siraz.

**Investigation:** M. M. Mahfuz Siraz.

**Methodology:** M. S. Alam, M. M. Mahfuz Siraz, Jubair A. M.

**Project administration:** M. M. Mahfuz Siraz.

**Supervision:** M. M. Mahfuz Siraz, Mayeen Uddin Khandaker, Afroza Shelley, Selina Yeasmin.

**Validation:** M. M. Mahfuz Siraz.

**Visualization:** M. M. Mahfuz Siraz.

**Writing – original draft:** M. S. Alam, M. M. Mahfuz Siraz, Jubair A. M., S. C. Das.

**Writing – review & editing:** M. S. Alam, Jubair A. M., D. A. Bradley, Mayeen Uddin Khandaker, Shinji Tokonami.

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
