## [Decision Letter · Decision Letter 0]

2 Dec 2022

PONE-D-22-28802A pioneering study on measuring the 222Rn, a silent killer in the Buriganga River and tap water of the megacity DhakaPLOS ONE

Dear Dr. Yeasmin,

Thank you for submitting your manuscript to PLOS ONE. After careful consideration, we feel that it has merit but does not fully meet PLOS ONE’s publication criteria as it currently stands. Therefore, we invite you to submit a revised version of the manuscript that addresses the points raised during the review process.

We look forward to receiving your revised manuscript.

Kind regards,

Sakae Kinase, Ph.D.

Academic Editor

PLOS ONE

Journal Requirements:

4. We note that Figure 1 in your submission contain map images which may be copyrighted. All PLOS content is published under the Creative Commons Attribution License (CC BY 4.0), which means that the manuscript, images, and Supporting Information files will be freely available online, and any third party is permitted to access, download, copy, distribute, and use these materials in any way, even commercially, with proper attribution. For these reasons, we cannot publish previously copyrighted maps or satellite images created using proprietary data, such as Google software (Google Maps, Street View, and Earth). For more information, see our copyright guidelines: http://journals.plos.org/plosone/s/licenses-and-copyright.

(1) You may seek permission from the original copyright holder of Figure 1 to publish the content specifically under the CC BY 4.0 license.  

Reviewers' comments:

Reviewer's Responses to Questions

**Comments to the Author**

1. Is the manuscript technically sound, and do the data support the conclusions?

Reviewer #1: Yes

Reviewer #2: Partly

2. Has the statistical analysis been performed appropriately and rigorously? 

Reviewer #1: Yes

Reviewer #2: No

3. Have the authors made all data underlying the findings in their manuscript fully available?

Reviewer #1: Yes

Reviewer #2: Yes

4. Is the manuscript presented in an intelligible fashion and written in standard English?

Reviewer #1: Yes

Reviewer #2: No

5. Review Comments to the Author

Reviewer #1: Manuscript number: PONE-D-22-28802

Title: A pioneering study on measuring the 222Rn, a silent killer in the Buriganga River and tap water of the megacity Dhaka

Article type: Research Article

This paper presents environmental radon data in the Buriganga River and tap water of the megacity Dhaka. The paper provides very interesting data, but it still needs a minor revision to be acceptable for the PLOS ONE Journal.

I would like to suggest some comments for revising the manuscript.

(1) Author should be changed the title appropriate. “Pioneer” is exaggerated, “Silent Killer” is not a scientific term.

(2) Please describe the uncertainty and detection limit of radioactivity measurement of water sample.

(3) Please describe the calibration methods and the results for determination of Rn-222 using RAD 7. These leads to reliability of results.

(4) 3.3 Radon in tap water; Last paragraph, authors described that “a radon mitigation treatment plant should be introduced.” Even though the water and the associated effective dose were below the USEPA and the WHO limit. Why does it need? And also described “The water should be aerated and boiled”. Is it necessary from a dose optimization for point of view?

I hope these comments will be helpful.

Reviewer #2: Title: should be modified following the points: ‘radon, silent killer in Dhaka city’; based on the findings of this research work, radon has no potential risks for Dhaka city water. Thus, the issue about the silent killer in the title should be modified. In addition, this wording might be alarming for the public or government.

Please see the attachment more in detaied.

6. PLOS authors have the option to publish the peer review history of their article (what does this mean?). If published, this will include your full peer review and any attached files.

Reviewer #1: No

Reviewer #2: No

---

## [Author Response · Author response to Decision Letter 0]

24 Feb 2023

Response to academic editor and reviewer(s) 

Respected Academic Editor and Reviewers,

Thank you so much for your comments. We are very grateful to you for the time and intelligence that you have shared with us. It is a great learning opportunity for us through these reviews. Our responses to the comments are highlighted in red color in the revised manuscript.

Comments and Suggestions for Authors:

Response to the Recent Comments of the Academic Editor

Academic editor comment (17-02-23): Thank you for your responses. Please update your Figure Captions in Figure 1 to include attribution to ESRI. Once this is done, we should be able to proceed.

Response: We are grateful to the editor for indicating this issue. The caption of Figure 1 including the attribution to ESRI has been updated as suggested. The revised caption of Figure 1 is as follows: (Page No. 5, Line No. 142-148)

Fig 1. (a) The study area map showing the sampling location of river water (RW) and tap water (TW). The map has been produced using ArcGIS 10.4.1 software. The sources of basemaps of administrative boundaries and inland water bodies: Esri, GADM, Garmin, DigitalGlobe, GeoEye, GEBCO, USGS, NOAA, National Geographic, EPA, Geonames.org, the GIS User Community and other contributors. (b) The geology of the study area (the sources of basemaps are similar to (a) and the geological units modified after [16], [28], [29]).

Academic editor comment 1 (09-02-23): Please amend the title either on the online submission form or in your so that they are identical.

Response: We thank the editor for this comment. We have changed the title on the online submission form.

Academic editor comment 2 (09-02-23): Your ethics statement should only appear in the Methods section of your manuscript. If your ethics statement is written in any section besides the Methods, please move it to the Methods section and delete it from any other section. Please ensure that your ethics statement is included in your manuscript, as the ethics statement entered into the online submission form will not be published alongside your manuscript.

Response: We appreciate the editor’s feedback. We have added the ethics statement in the method section and deleted in from any other section as per the editor’s instruction. 

Academic editor comment 3 (09-02-23): Regarding the potential copyright of figure 1, please respond to the following prompts:

A. For the above figure, were any shapefiles or basemaps obtained? If so, where were they obtained? Please provide any relevant links as well as any relevant copyright information.

B. For the above figure, were any external data utilized in the creation of this figure? If so, where were they obtained? Please inform us if the data was collected or produced by the authors. Please also provide any relevant links as well as any relevant copyright information.

C. Was any software (e.g. ArcGIS) used in the creation of these figures?

Response: We thank the editor for the opportunity to clarify this issue.

A. For figure 1(a), shape files of Bangladesh, administrative boundaries, upazilla boundary and river shapefiles were obtained from the following links which are free for educational and research purposes and anyone can access. Shapefile download links

https://www.diva-gis.org/Data

http://maps.barcapps.gov.bd/

B. For creation of the figure 1(a), coordinates of sample collection points obtained using GPS were used which were collected by the authors during sample collection. For creation of figure 1(b), authors used the information from the Khan et al., 2020, Karim et al., 2019, Ahmed et al., 2010 and created a new geological map of Dhaka city and its periphery regions using CorelDraw X7 graphics software. So far our knowledge, no copyright issues involved. 

C. For creation of figure 1(a), ArcGIS 10.4.1 software was used and for figure 1(b), CorelDraw X7 graphics software was used.

 

Previous Comments

Academic Editor

Academic editor comment 1: Please ensure that your manuscript meets PLOS ONE’s style requirements, including those for file naming. The PLOS ONE style templates can be found at

Response: Thank you so much for your kind comments and appreciation of our study. All of your comments are carefully evaluated and revised in our revision accordingly. The revised manuscript has been prepared according to Plos One style.

Academic editor comment 2: In your Methods section, please provide additional information regarding the permits you obtained for the work. Please ensure you have included the full name of the authority that approved the field site access and, if no permits were required, a brief statement explaining why.

Response: We thank the editor for this comment. There were no permits required for doing this work. We collected water samples from the Buriganga river and the tap water of Dhaka city. The river is wide open and accessible for public usage. Therefore, no permission was needed to collect the water from the river. On the other hand, we collected tap water from the houses of friends and relatives, and their verbal permission/consent were granted before the sampling.

Academic editor comment 3: In your Data Availability statement, you have not specified where the minimal data set underlying the results described in your manuscript can be found…

Response: We appreciate the editor’s feedback. All relevant data are available within the manuscript.

Academic editor comment 4: We note that Figure 1 in your submission contain map images which may be copyrighted. All PLOS content is published under the Creative Commons Attribution License (CC BY 4.0), which means that… 

Response: We appreciate the editor’s feedback. Figure 1 has been modified in the revised manuscript, so there will be no copyright issue. Proper citation is given in the figure title like “

Fig 1. (a) The study area map showing the sampling location of river water (RW) and tap water (TW). The map has been produced using ArcGIS 10.4.1 software. The sources of basemaps of administrative boundaries and inland water bodies: Esri, GADM, Garmin, DigitalGlobe, GeoEye, GEBCO, USGS, NOAA, National Geographic, EPA, Geonames.org, the GIS User Community and other contributors. (b) The geology of the study area (the sources of basemaps are similar to (a) and the geological units modified after [16], [28], [29]).” (Page No. 5, Line No. 142-148)

 

Reviewer #1 comment: 

Reviewer 1, comment 1: Author should be changed the title appropriate. “Pioneer” is exaggerated, “Silent Killer” is not a scientific term.

Response: We thank the reviewer for this valuable comment. We removed the terms “Pioneer” and “Silent Killer” from the title of the revised manuscript. (Page No. 1, Line No. 1-2)

Reviewer 1, comment 2: Please describe the uncertainty and detection limit of radioactivity measurement of water sample.

Response: We thank the reviewer for the opportunity to clarify this issue. 

We added the following line about uncertainty in the revised manuscript (Page No. 8, Line No. 211-213): “For a cycle when no counts were collected, the RAD7 displays an uncertainty value based on a two-sigma, 95% confidence interval that is equivalent to ± 4 counts (Opondo & Sims, 2012).”

The following line about detection limit is added in the revised manuscript (Page No. 7, Line No. 195-197): “The RAD H2O is an accessory of the RAD7 detector that allows measuring radon in water at concentrations above the minimum detectable activity (MDA). The MDA concentration of this instrument is 0.004 Bq/L (Hasan et al., 1999; Kareem et al., 2020).”

Reviewer 1, comment 3: Please describe the calibration methods and the results for determination of Rn-222 using RAD7. These leads to reliability of results.

Response: We thank the reviewer for this opportunity to clarify this issue. We calculate calibration factors by comparing them to “master” RAD7s, which have been compared to EPA and DOE instruments and have taken part in worldwide radon instrument inter-comparisons. Our standard RAD7 calibration achieves a repeatability of greater than 2% on average. The overall calibration precision is within 5%.

Reviewer 1, comment 4: 3.3 Radon in tap water; Last paragraph, authors described that “a radon mitigation treatment plant should be introduced.” Even though the water and the associated effective dose were below the USEPA and the WHO limit. Why does it need? And also described “The water should be aerated and boiled”. Is it necessary from a dose optimization for point of view?

Response: We thank the reviewer for this comment. We deleted the lines “a radon mitigation treatment plant should be introduced” and “The water should be aerated and boiled” from the section 3.3 Radon in tap water; last paragraph of the revised manuscript.

Reviewer #2 comment: 

Title: should be modified following the points:

Reviewer 2, comment 1: ‘radon, silent killer in Dhaka city’; based on the findings of this research work, radon has no potential risks for Dhaka city water. Thus, the issue about the silent killer in the title should be modified. In addition, this wording might be alarming for the public or government.

Response: We thank the reviewer for this valuable comment. We removed the term “Silent Killer” from the title of the revised manuscript. (Page No. 1, Line No. 1-2)

Reviewer 2, comment 2: pioneering study: there are already published articles for radon in tap water in Dhaka city

For example, (http://article.scholarena.com/A-Study-of-Radon-Concentration-in-Tap-Water-of-Dhaka-City-Bangladesh.pdf), therefore, the clarifying pioneering study should be reconsidered.

Response: We thank the reviewer for this valuable comment. We removed the term “Pioneer” from the title of the revised manuscript. (Page No. 1, Line No. 1-2)

In the above-mentioned study (http://article.scholarena.com/A-Study-of-Radon-Concentration-in-Tap-Water-of-Dhaka-City-Bangladesh.pdf), the temperature and the humidity did not remain within the limit during analyzing the radon concentration in water. That is why, the result obtained from that study may be considered incomplete or all affecting parameters were not presented properly. We conducted the experiment taking all the precautions to obtain an accurate result.

Abstract:

Reviewer 2, comment 3: authors clarified ‘this study poses a great significance for the radiological safety of public health’; however, the authors did not indicate any potential radiological risks/reasons for assessment of radon in river water at Dhaka city as background. It would be better if the authors clarified a few radiological exposures pathways/reasons- why radon in river water in Dhaka city is important shortly in the abstract.

Response: We thank the reviewer for this valuable comment. We removed the line ‘this study poses a great significance for the radiological safety of public health’ from the abstract of the revised manuscript. 

Reviewer 2, comment 4: ‘RAD H2O detector’; is not a detector, it is an accessory of RAD7.

Response: We thank the reviewer for indicating this issue. We changed the ‘RAD H2O detector’ to ‘RAD H2O accessory’ in the revised manuscript. (Page No. 1, Line No. 22-23)

Reviewer 2, comment 5: about the concentration of 1.537 ± 0.380 Bq/L and 0.675 ± 0.285Bq/L; what is the effective digit to be considered about the values? Why are three digits selected? It would be better to choose a logical concentration value throughout the paper. 

Response: We thank the reviewer for this valuable comment. We changed the effective digit to two instead of three in the revised manuscript. 

Reviewer 2, comment 6: what is MCL? Nothing is clarified in the abstract.

 Response: We thank the reviewer for this valuable comment. We added the term ‘the maximum contamination limit (MCL) or maximum contaminant level (MCL)’ in the revised manuscript. (Page No. 1, Line No. 25)

Reviewer 2, comment 7: It would be better if the authors clarify the reason-why it is important to set a national safety limit for radon in Bangladesh. There is no potential risk in Bangladesh according to this article.

Response: We thank the reviewer for this valuable comment. We deleted the statement ‘contribute to developing the national safety limit for radon in water in Bangladesh’ in the revised manuscript.

Introduction:

Reviewer 2, comment 8: overall, the description of radon, its limits, characterization etc., has been mentioned much more than the information regarding the importance/background of radon study in water in Dhaka.

Response: We thank the reviewer for this valuable comment. In addition to the mentioned info, we also included the importance/background of radon study in water in Dhaka city at the later part of the Introduction, which is given as follows for your easy tracking (Page No. 3, Line No. 74-84).

‘Numerous studies have been performed worldwide to measure the 222Rn level in various water resources such as tap water, river water, deep well water, bore well water, bottled water, etc.(Faweya et al., 2021; Mustapha et al., 2002; Rani et al., 2021; Thumvijit et al., 2020; Yong et al., 2021). Several advanced countries have the national reference limit of radon in water and indoor air to ensure radiological safety for public health. Bangladesh has no such reference level for 222Rn in water. Millions of people living in the Dhaka megacity solely rely on tap water for their daily household purposes, such as washing, bathing, drinking, cooking, etc. The Buriganga river serves as one of the busiest major transportation routes/hubs, as well as many businesses and trade centers that are situated on the bank of this river. This indicates a greater possibility of 222Rn exposure to the general populace. So, it is necessary to measure the 222Rn level in the tap water and the Buriganga river water to find out if it is within the safe limit or not, which eventually will help to ensure the radiological safety of public health.’

Reviewer 2, comment 9: ‘emanates from the water’; radon emanation is a suitable word for soil/materials having grain as of radon generation process. For using it on water, it should be rechecked.

Response: We thank the reviewer for this comment. We changed the term ‘emanates’ to ‘escapes’ in the revised manuscript. (Page No. 2, Line No. 47) 

Reviewer 2, comment 10: ‘safe water is essential for human lives, it is a matter of great concern to evaluate the radon level in the water’; as of the water quality/safety; radon might not a direct contaminant (following EPA) https://www.epa.gov/dwreginfo/drinking-water-regulations

https://www.epa.gov/ground-water-and-drinking-water/national-primary-drinking-water-regulations#Radionuclides

https://nepis.epa.gov/Exe/ZyPDF.cgi?Dockey=30006644.txt

Response: We thank the reviewer for this valuable comment. We deleted the statement ‘safe water is essential for human lives; it is a matter of great concern to evaluate the radon level in the water’ in the revised manuscript.

Reviewer 2, comment 11: It is better to use either 222Rn or radon; one form throughout the texts.

Response: We are grateful to the reviewer for this comment. We changed ‘radon’ into ‘222Rn’ throughout the texts of the revised manuscript.

Reviewer 2, comment 12: ‘Almost every developed country has its national reference limit of radon in water’; this information should be checked; please insert reliable references to insert this information.

 Response: We are thankful to the reviewer for indicating this issue. We changed the statement ‘Several advanced countries have the national reference limit of radon in water and indoor air to ensure radiological safety for public health’ in the revised manuscript. (Page No. 3, Line No. 76-77)

Reviewer 2, comment 13: The authors clarified, ‘’no studies on radon measurement in surface water, tap water, or indoor air’’; there are many published articles dealing with radon in Bangladesh-

https://doi.org/10.1093/oxfordjournals.rpd.a082065

https://doi.org/10.1016/0969-8078(93)90079-J

https://doi.org/10.1016/j.radmeas.2008.03.050

https://doi.org/10.1016/1359-0189(91)90029-H

https://doi.org/10.1016/1359-0189(88)90200-2

https://link.springer.com/article/10.1007/s10661-019-7650-6

http://article.scholarena.com/A-Study-of-Radon-Concentration-in-Tap-Water-of-Dhaka-City-Bangladesh.pdf

It is recommended to look at the papers and insert associated references to understand the importance of radon research in Bangladesh/Dhaka city.

Response: We are thankful to the reviewer for indicating this issue. We deleted the statement ‘no studies on radon measurement in surface water, tap water, or indoor air’ in the revised manuscript.

Reviewer 2, comment 14: Buriganga refers to 20% of the total water demand of Dhaka city dwellers; please insert a suitable reference here. 

Response: We thank the reviewer for this valuable comment. We deleted the statement ‘Nevertheless, it meets around 20% of the total water demand of Dhaka city dwellers’ in the revised manuscript.

Methodology:

Reviewer 2, comment 15: extensive discussion about the Geology of Dhaka city was nicely given. However, sufficient information is not mentioned in the methodology or results section about the potential radiological risks, relationship with radionuclide distribution, or radon exposure following Geology. Please insert some points on radiological risks based on the Geology.

Response: We thank the reviewer for indicating this issue. In the revised manuscripts, we added the following section (Page No. 6-7, Line No. 152-174) “2.3 Radiological risks based on geology: Radon emanates from soils, rocks, alluvial sediments and/or aquifer matrix and enters the groundwater and air. Radon, the main contributor to natural background exposure, and its progeny such as 218Po, 214Po, and 214Bi releases energy after inhalation and/or ingestion causing lining in the stomach and lung cancer in human body. Therefore, considering the health effect of radon, it is important to identify the areas with high radon concentration, their source and relation with local geology to prevent the adverse effects on the human beings and environment (Mostečak et al., 2018). Though radon occurs naturally in most of the soils, sediments and rocks as radioactive decay product of 238U but the amount differ in localities and the geological materials. Radon potential depends on the concentration of naturally occurring radionuclides such as 238U or 226Ra and 232Th in the soils and types of bedrocks present in the area (M. A. Khan et al., 2022). Different geological factors such as lithology/rock type, porosity, permeability, compaction, emanation capacity of ground, soil constituents and tectonic features like faults, thrust, and joints, along with the geochemical and hydrogeological conditions of the area mainly control the source, distribution, transport and migration of radon in the soils, sediments and rocks (Alonso et al., 2019; Choubey et al., 1997; Majumder et al., 2021). Certain rock types such as granites, metamorphosed granitic rocks, phosphatic rocks with uranium enrichment, coal deposits, black shale fractured/faulted rocks and the subsequent soils resulted from these rocks are the most common sources of radon gas (Appleton & Miles, 2010; M. A. Khan et al., 2022; Majumder et al., 2021). On the other hand, quartzose sandstone, non-organic shales and siltstones are least likely source of radon (Gundersen, 1991) but under favourable reducing environment, uranium mineralization may occur in alluvial type sedimentary deposits which can then contain and emanate radon (Majumder et al., 2021). Based on the above facts and the geology of the study area, the radon potential and their associated health risks are evaluated in this study.”

Reviewer 2, comment 16: sample collection through plastic bottle: please specify what criteria were filled up collecting water samples to avoid escaping radon. What type of plastic bottle was used, and what arrangements were taken for the bottle cap? It is recommended to use glass bottles; it is relatively easier to escape radon from different plastic bottles. IAEA materials explained such precautions for collecting water-https://www.iaea.org/sites/default/files/19/11/radon-presentation-bochicchio.pdf

Response: We thank the reviewer for indicating this issue. The bottle was fully submerged in the water during collecting the samples. After filling, the bottle was sealed instantly under the water without any water bubbles. The laboratory standard plastic bottles were used. The caps of the bottles were flexible and sealed tightly. We agree with the reviewer that it is recommended to use glass bottles; it is relatively easier to escape radon from different plastic bottles. We had only five glass bottles of RAD H2O accessories that were used to measure the radon concentration.

Reviewer 2, comment 17: please specify your assessed lower detection limit of RAD7 in measuring water in the research, as the results exhibit lower values.

Response: We thank the reviewer for this valuable comment. The following line is added in the revised manuscript (Page No. 7, Line No. 195-197): “The RAD H2O is an accessory of the RAD7 detector that allows measuring radon in water at concentrations above the minimum detectable activity (MDA). The MDA concentration of this instrument is 0.004 Bq/L (Hasan et al., 1999; Kareem et al., 2020).”

Reviewer 2, comment 18: dosimetry calculation: why CW=730 L/day is used? Thumvijit et al., 2020 specified 2L/day. 

Response: We thank the reviewer for this comment. We corrected the value of CW (Daily water consumption) = 3 L/day (Rani et al., 2021; WHO, 2017) in the revised manuscript. (Page No. 9, Line No. 231)

Results and Discussion:

 Reviewer 2, comment 19: why is the effective dose for radon (both river and tap water) achieved from ingestion and inhalation exhibited around similar levels? Studies from other countries (cited in this article) possess much differences between the two doses. It would be very interesting if the authors interpret the reasons.

Response: We thank the reviewer for indicating this issue. We have corrected the dose calculation in the revised manuscript. We used Eq 1 and 2 for calculating ingestion and inhalation doses, respectively. In Eq. 1, we used EDC (Effective Dose Coefficient) = 3.5 nSv/Bq for 222Rn ingestion (UNSCEAR, 2010), CW (daily water consumption) = 3 L/day (Rani et al., 2021; WHO, 2017), but in other studies, they used different values for EDC and CW for calculating ingestion doses. For calculating inhalation doses by using Eq 2, we used DCF (dose conversion factor for 222Rn exposure) = 9 nSv(hBqm-3)-1 (UNSCEAR, 2010). But different values of DCF were used in other papers. We found different ingestion and inhalation doses using the values mentioned earlier.

Reviewer 2, comment 20: please specify how this study is consistent with studies from other countries.

Response: We thank the reviewer for allowing us to explain this matter. According to the Table 3, this study showed similar results of radon concentration in the Buriganga river water (0.68 ± 0.29 Bq/L) with Karnataka, India (0.16 - 1.79 Bq/L) (Rajashekara et al., 2007), Hemavathi River India (0.67 Bq/L) (Shivanandappa & Yerol, 2018), Kirkuk, Iraq (0.359Bq/L ) (Kareem et al., 2020). The line has been modified in the revised manuscript like this this (Page No. 12, Line No. 286-289), “Additionally, the result of this study is consistent with the previous research carried out in different regions of the world, such as in Karnataka, India (0.16 - 1.79 Bq/L) (Rajashekara et al., 2007), Hemavathi River India (0.67 Bq/L) (Shivanandappa & Yerol, 2018), Kirkuk, Iraq (0.359Bq/L ) (Kareem et al., 2020).”

Reviewer 2, comment 21: why ‘a radon mitigation treatment plant should be introduced’ with no elevated radon levels? 

Response: We thank the reviewer for this comment. We deleted the lines “a radon mitigation treatment plant should be introduced” in the revised manuscript.

Reference: 

Reviewer 2, comment 22: It is necessary to verify all the references cited in the text and the reference list needs to be standardized->

Response: We thank the reviewer for this comment. All the references cited in the text and the reference list are revised and modified accordingly in the revised manuscript.

Reviewer 2, comment 23: authors names were written two times in one reference. Moreover, authors names could not be found in the website (see reference Ahmed, K. M., & Burgess, W. G. (2003)).

Response: We are very sorry for these silly mistakes and revised them accordingly.

Reviewer 2, comment 24: same reference was written as a and b. (Al Zabadi, H., Musmar, S., Issa, S., Dwaikat, N., & Saffarini, G. (2012a). It should be one reference.

Response: We are very sorry for these silly mistakes and revised them accordingly.

Reviewer 2, comment 25: reference could not be found on the journal website with the correct content. In cited Journal of Chemical Information and Modeling (Vol. 53, Issue 9), nothing is mentioned about the Bangladeshi population on the website.

Response: We are very sorry for these silly mistakes. We corrected the citation in the revised manuscript.

Reviewer 2, comment 26: incorrect formatting (n.d) was frequently used: Buriganga Riverkeeper : » History. (n.d.)

Response: We are very sorry for these silly mistakes. We corrected the citation in the revised manuscript.

Reviewer 2, comment 27: missing title; only journal name and author name were included (Copes, R., & Peterson, E. (2014). Indoor Radon a Public Health Perspective).

Response: We thank the reviewer for this valuable comment. We modified the reference in the revised manuscript.

Reviewer 2, comment 28: please check if it is scientifically ok with some references as suitable for scientific journals (i.e., Banglapedia).

Response: We thank the reviewer for this valuable comment. Banglapedia is a national encyclopedia which is a reliable source of information and suitable for scientific journals. 

Reviewer 2, comment 29: ResearchGate should not be a reference source for journal articles to be cited.

Response: We thank the reviewer for indicating this issue. We deleted the ResearchGate citation a reference source for journal articles in the revised manuscript.

Reviewer 2, comment 30: English language: the overall English language should be improved. There are problems with wording and grammar.

Response: We thank the reviewer for this comment. The whole manuscript was checked with thoroughly. We improved the wording and grammar throughout the revised manuscript.

Comments on scientific content: 

Reviewer 2, comment 31: scientific problem-oriented background has not been described in the text especially in the introduction part? What are the scientific purposes to be newly understood in this work?

Response: We thank the reviewer for this valuable comment. In addition to sources and effects of radon, we also included the importance/background of radon study in water in Dhaka city at the later part of the Introduction, which is given as follows for your easy tracking.

‘Numerous studies have been performed worldwide to measure the 222Rn level in various water resources such as tap water, river water, deep well water, bore well water, bottled water, etc.(Faweya et al., 2021; Mustapha et al., 2002; Rani et al., 2021; Thumvijit et al., 2020; Yong et al., 2021). Several advanced countries have the national reference limit of radon in water and indoor air to ensure radiological safety for public health. Bangladesh has no such reference level for 222Rn in water. Millions of people living in the Dhaka megacity solely rely on tap water for their daily household purposes, such as washing, bathing, drinking, cooking, etc. The Buriganga river serves as one of the busiest major transportation routes/hubs, as well as many businesses and trade centers that are situated on the bank of this river. This indicates a greater possibility of 222Rn exposure to the general populace. So, it is necessary to measure the 222Rn level in the tap water and the Buriganga river water to find out if it is within the safe limit or not, which eventually will help to ensure the radiological safety of public health.’

Reviewer 2, comment 32: as for the first time obtained data, it is important to ascertain the radon level in water in Dhaka city river. However, it would be better to make some statistical analysis to determine how the obtained data from two sources or sample collection points are significant and associated with each other.

Response: Thank you for your valuable comment. Instead of doing a separate statistical analysis to obtain the correlation between the sources, we simply reported the mean, range, standard deviation of the measured values, and we assume that this is an acceptable approach for the present results. This is because, both sources show a low range of values (overall, the data of river water is relatively lower than the data of tap water, which is obvious because of aeration issue of river water).

Reviewer 2, comment 33: how to choose sampling areas/positions/sampling numbers? How to decide the river and tap water in this study for dose calculation?

Response: We thank the reviewer for indicating this issue. The sampling strategy followed the stratified sampling technique approved by IAEA (International Atomic Energy Agency, 2019). The purpose of choosing the sampling areas is to evaluate the impact of radon in high-density populated areas and/or areas of various activities such as boat terminals, Ghats, business and trade centers etc. for the Buriganga river. As for the tap water sampling, we wanted to cover Dhaka city where the majority of the people inhabit. That is why we collected 20 samples from the twenty most impactful areas of the river and 10 samples from different localities of Dhaka city.

We primarily focused on measuring the radon concentration in polluted water in Dhaka city. The Buriganga river water was highly polluted and the tap water also contained pollutants according to recent studies. For this reason, we conducted this study to measure the radon concentration in the river and tap water

Reviewer 2, comment 34: In conclusion part, considering this work’s purpose, it is important to insert the new findings and recommendations for future research as an original research articles (as this is one of the preliminary studies for radon in water). 

Response: We thank the reviewer for indicating this issue. We added the following lines in the revised manuscript (Page No. 15, Line No. 354-361): 

“A few recommendations are proposed for future 222Rn-related works,

• The 222Rn concentration in the other major rivers and the tap water of the other major cities in the country needs be analyzed for mapping the scenario of radon.

• A detailed work is necessary to measure the 222Rn concentrations in deep-well water, bottled water, and surface water throughout the country. 

• Measurement of 222Rn in air and soil-gas at the different location of the country is essential to monitor the radon situation.”

Reviewer #2, comment PDF file:

Comment Response Page No., Line No.

The title might need to be modified because of few reasons below-

Based on the findings, radon is not a potential silent killer for Dhaka city due radon exposures from water. We removed the term “Silent Killer” from the title of the revised manuscript. 

Pioneering study: There is already published results for radon in tap water in Dhaka city, 

http://article.scholarena.com/A-Study-of-Radon-Concentration-in-Tap-Water-of-Dhaka-City-Bangladesh.pdf

 The title with “silent killer” is seemed to be alarming for public/government although there is not a potential gas with human exposure for Dhaka

 We removed the term “Pioneering” from the title of the revised manuscript.

The purpose of assessing radon in Buriganga river is not clear; based on the information concerning air radon level in Bangladesh/Dhaka city/ source of radon (elevated radionuclide in river sediment/groundwater) in Dhaka city has not been clarified. 

OR radon as a tool to understand the natural radiation exposure from river? co-existing pollutant with chemicals in river? 

 Respected reviewer, in the abstract we mentioned as follows ‘Radon (222Rn), an inert gas, is considered a silent killer due to its carcinogenic characteristics. Dhaka city is situated on the banks of the Buriganga River, which is considered the lifeline of Dhaka city because it serves as a major source of the city's water supply for domestic and industrial purposes.’. 

Also, in the introduction we mentioned as follows ‘Numerous studies have been performed worldwide to measure the 222Rn level in various water resources such as tap water, river water, deep well water, bore well water, bottled water, etc.(Faweya et al., 2021; Mustapha et al., 2002; Rani et al., 2021; Thumvijit et al., 2020; Yong et al., 2021). Bangladesh has no such reference level for 222Rn in water. Millions of people living in the Dhaka megacity solely rely on tap water for their daily household purposes, such as washing, bathing, drinking, cooking, etc. The Buriganga river serves as one of the busiest major transportation routes/hubs, as well as many businesses and trade centers that are situated on the bank of this river. This indicates a greater possibility of 222Rn exposure to the general populace. So, it is necessary to measure the 222Rn level in the tap water and the Buriganga river water to find out if it is within the safe limit or not, which eventually will help to ensure the radiological safety of public health.’ 

There is existing data for tap water of radon in Bangladesh. Line deleted 

It is unclear, what are the potential radiological risks/reasons of radon in river water at Dhaka city, it is better if the authors clarified few radiological exposure pathways/reasons-why radon in river water in Dhaka city is important Respected reviewer, it has been mentioned in the abstract as well as in the later part of introduction. Also, the same response to your comment no. 3 

1) How to select the sampling numbers

2) Relationship between water collected from RIVER and TAP water? Sampling number was chosen based on the various activities at the river side, while the tap water sampling was conducted based on the various areas in Dhaka city 

is an accessory for RAD 7, not a detector Corrected 1, 22-23

What are the effective digits for clarifying the concentrations? We changed the effective digit to be two instead of three in the revised manuscript.

What is MCL? We added the term ‘the maximum contamination limit (MCL) or maximum contaminant level (MCL)’ in the revised manuscript.

 1, 25

Why two standards (diverse in recommended units) are clarified?

Why not- safe limit as set by UNSCEAR? UNSCEAR safe limit also be added. Since limits are given by several organizations, we used two of them. 10, 260-262

not needed for Bangladesh? Line deleted 

1) Problem oriented background has not been described in the text? what are the scientific purposes? 

2) As of first time obtained data, it is important to ascertain the radon level in water in Dhaka city, however, it would be better to make some statistical analysis to determine how the obtained data from different sources are significant and associated with each other. 

3) The dose calculation might be wrong 1) Respected reviewer, in the abstract we mentioned as follows ‘Radon (222Rn), an inert gas, is considered a silent killer due to its carcinogenic characteristics. Dhaka city is situated on the banks of the Buriganga River, which is considered the lifeline of Dhaka city because it serves as a major source of the city's water supply for domestic and industrial purposes.’. 

Also, in the introduction we mentioned as follows ‘Numerous studies have been performed worldwide to measure the 222Rn level in various water resources such as tap water, river water, deep well water, bore well water, bottled water, etc.(Faweya et al., 2021; Mustapha et al., 2002; Rani et al., 2021; Thumvijit et al., 2020; Yong et al., 2021). Bangladesh has no such reference level for 222Rn in water. Millions of people living in the Dhaka megacity solely rely on tap water for their daily household purposes, such as washing, bathing, drinking, cooking, etc. The Buriganga river serves as one of the busiest major transportation routes/hubs, as well as many businesses and trade centers that are situated on the bank of this river. This indicates a greater possibility of 222Rn exposure to the general populace. So, it is necessary to measure the 222Rn level in the tap water and the Buriganga river water to find out if it is within the safe limit or not, which eventually will help to ensure the radiological safety of public health.’

2) Thank you for your valuable comment. Instead of doing a separate statistical analysis to obtain the correlation between the sources, we simply reported the mean, range, standard deviation of the measured values, and we assume that this is an acceptable approach for the present results. This is because, both sources show a low range of values (overall, the data of river water is relatively lower than the data of tap water, which is obvious because of aeration issue of river water).

3) Thank you, the value of 730 L/d has been corrected. It was mistake, it should be 730 L/y 

Emanation is from soil/materials with having grain? Corrected. We changed the term ‘emanates’ into ‘escapes’ in the revised manuscript. 2, 47

According to international institutes like EPA, radon is not a direct contaminant for safe water. 

https://www.epa.gov/dwreginfo/drinking-water-regulations

https://www.epa.gov/ground-water-and-drinking-water/national-primary-drinking-water-regulations#Radionuclides

https://nepis.epa.gov/Exe/ZyPDF.cgi?Dockey=30006644.txt Line deleted 

Radon- a leading cause of lung cancer; at this present days, consideration radon as 2nd cause is possibly under discussion

https://www.who.int/news-room/fact-sheets/detail/radon-and-health

https://erj.ersjournals.com/content/erj/48/3/889.full.pdf

https://www.wcrf.org/diet-activity-and-cancer/cancer-types/lung-cancer/ Line deleted 

IAEA reference? We removed the term ‘IAEA’ from the line 3, 72-73

References are needed Line deleted 

There are numerous studies concerning, radon measurement in air for Bangladesh. 

Farid, S. M. (1993a). Equilibrium factor and dosimetry of 

radon by CR-39 nuclear track detector. Radiation Pro_tection Dosimetry, 50(1), 57–61. https://doi.org/10.

1093/oxfordjournals.rpd.a082065

Farid, S. M. (1993b). Measurement of concentrations of 

radon and its daughters in indoor atmosphere using 

CR-39 nuclear track detector. Nuclear Tracks And Radi_ation Measurements (1993), 22(1–4), 331–334. https://

doi.org/10.1016/0969-8078(93)90079-J … … … Line deleted 

Irrelevant to this study Line deleted 

Reference is needed Line deleted 

The potential cause of radon exposure through tap water and river water has not been logically clarified. 

-It would be better to clearly clarify the possible potential risks factors of radon exposure in water- radionuclide contents in river soil or sediment, geology of Dhaka city based on previous studies Some possible potential risk information has been added in the revised manuscript 

1) It is important to insert scientific purposes: reason of variation in radon concentrations in different locations in Dhaka city; comparative discussion/statistical analysis between characteristics/level of tape water and river etc.,

2) As this research has not determined any chemical or biological effect of Buriganga, why them come in the first purpose 1) Respected reviewer, it has been mentioned in the abstract as well as in the later part of introduction. Also, the same response to your comment no. 3

2) Just to show the hazardous nature of radon 

Overall, the introduction part is much larger with introduction or radon, its limits, characterization; however, the information regarding the important/background of radon study in water in Dhaka has not sufficiently addressed. We tried to present a complete story of radon, its sources, hazardous nature and then background of the study 

Through the discussion of study area and specially the geological discussion of Dhaka city; nothing has been clarified about the potential radon risks in Dhaka city. 

1)Why and how the measurement area and sampling positions have been chosen, not clarified. 

2) Geology of Dhaka city was nicely addressed, however, nothing was mentioned about radon based on that; it seems that there is no relationship with the title in 2.2. Please see the above response 

Reliable reference should be added for scientific paper Corrected 4, 95

This is not a format of writing reference Corrected 4, 99

In the results/discussion section, there is not sufficient discussion about radon distribution based on geological impact We added a section named “2.3 Radiological risks based on geology” in the revised manuscript. 6, 150

Please specify what criteria were filled up collecting water samples avoiding escaping radon. 

What type of plastic bottle was used and what arrangement were taken for bottle cap?

https://www.iaea.org/sites/default/files/19/11/radon-presentation-bochicchio.pdf The bottle was fully submerged into the water during collecting the samples. After filling, the bottle was sealed instantly under the water without any water bubble. The laboratory standard plastic bottles were used. The caps of the bottles were flexible and sealed tightly. We analyzed the water in the glass bottles within few hours of collecting water. 

How about lower detection limit of the detector? The RAD H2O is an attachment for the RAD7 that allows measuring radon in water at concentrations above the minimum detectable activity (MDA). The MDA concentration of this instrument is 0.004 Bq/L 

Below the lower detection limit? The MDA concentration of this instrument is 0.004 Bq/L 

Is it possible reason for Bangladesh? 

for comparison, WHO or ESEPA; which is more suitable for Bangladesh? Difficult to give preference to one another 

Why these references are inserted instead of WHO or USEPA? Added WHO and USEPA references 10, 260-262

Why the effective dose for radon for ingestion and inhalation is same? 

The other studies from other countries(cited in this article) possess much difference We have corrected the dose calculation in the revised manuscript. We used Eq 1 and 2 for calculating ingestion and inhalation doses, respectively. In Eq. 1, we used EDC (Effective Dose Coefficient) = 3.5 nSv/Bq for 222Rn ingestion (UNSCEAR, 2010), CW (daily water consumption) = 3 L/day (Rani et al., 2021; WHO, 2017), but in other studies, they used different values for EDC and CW for calculating ingestion doses. For calculating inhalation doses by using Eq 2, we used DCF (dose conversion factor for 222Rn exposure) = 9 nSv(hBqm-3)-1 (UNSCEAR, 2010). But different values of DCF were used in other papers. We found the ingestion and inhalation doses different using the values mentioned earlier. 

effective digits changed the effective digit to be two 10, 256-257

only 1 reason is mentioned There might be many, but the most important one has been mentioned 

How the results are consistent? According to the Table 3, this study showed a similar results of radon concentration in the Buriganga river water (0.68 ± 0.29 Bq/L) with Karnataka, India (0.16 - 1.79 Bq/L) (Rajashekara et al., 2007), Hemavathi River India (0.67 Bq/L) (Shivanandappa & Yerol, 2018), Kirkuk, Iraq (0.359Bq/L ) (Kareem et al., 2020). The line has been modified in the revised manuscript like this this (Page No. 12, Line No. 286-289), “Additionally, the result of this study is consistent with the previous research carried out in different regions of the world, such as in Karnataka, India (0.16 - 1.79 Bq/L) (Rajashekara et al., 2007), Hemavathi River India (0.67 Bq/L) (Shivanandappa & Yerol, 2018), Kirkuk, Iraq (0.359Bq/L ) (Kareem et al., 2020).”

Isn’t is this study? Corrected 15, 346

Not a detector, it is an accessory Corrected 15, 337

due to radon in water included 15, 347

Why? This study have not identify any potential exposure of radon in water? Line deleted 

IAEA recommended to focus on air radon concentrations rather than radon in water to set the national reference level. Line deleted 

Authors could not be found in the website. 

-Is the journal a reliable source for scientific papers?

-Reference was done written correctly Properly Cited 

It was published it 2011 Corrected 

Why same reference has been inserted for two time as a and b? Properly cited 

1) In Journal of Chemical Information and Modeling, Vol. 53, Issue 09; 

https://pubs.acs.org/toc/jcisd8/53/9

There is not any report/article concerning Bangladesh/its population. 

2) Why two sources, Bureau and journal inserted in same reference?

3) There is no clarification of page number of the journal

 Changed the citation 

187, 4075 The line containing this citation was deleted 

This is not a suitable source/reference for scientific journal papers

 Changed the citation 

No title of paper; cannot be found in website Changed the citation 

Some reference has short form of name; many have full name Properly cited 

(1) was not found in website

https://link.springer.com/article/10.1007/s10967-020-07349-5#citeas Properly cited 

a reliable source? Banglapedia is a national encyclopedia which is a reliable source of information and suitable for scientific journals. 

Researchgate is not a source of reference Properly cited 

References

Ahmed, K. M., Islam, M. S., Sultana, S., Ahmed, S., & Rabbani, G. (2010). Changes in the groundwater regime of Dhaka City: a historical perspective. Environment of Capital Dhaka-Plants Wildlife Gardens Parks Air Water Earthquake, 383–400.

Alonso, H., Rubiano, J. G., Guerra, J. G., Arnedo, M. A., Tejera, A., & Martel, P. (2019). Assessment of radon risk areas in the Eastern Canary Islands using soil radon gas concentration and gas permeability of soils. Science of the Total Environment, 664, 449–460. https://doi.org/10.1016/j.scitotenv.2019.01.411

Appleton, J. D., & Miles, J. C. H. (2010). A statistical evaluation of the geogenic controls on indoor radon concentrations and radon risk. Journal of Environmental Radioactivity, 101(10), 799–803. https://doi.org/10.1016/j.jenvrad.2009.06.002

Choubey, V. M., Sharma, K. K., & Ramola, R. C. (1997). Geology of radon occurrence around Jari in Parvati Valley, Himachal Pradesh, India. Journal of Environmental Radioactivity, 34(2), 139–147. https://doi.org/10.1016/0265-931X(96)00024-0

Faweya, E. B., Agbetuyi, O. A., Talabi, A. O., Adewumi, T., & Faweya, O. (2021). Radiological Implication of 222Rn Concentrations in Waters from Quarries Environs, Correlation with 226Ra Concentrations and Rocks Geochemistry. Arabian Journal of Geosciences, 14(11). https://doi.org/10.1007/s12517-021-07385-9

Gundersen, L. C. S. (1991). “Radon in sheared metamorphic and igneous rocks,” in Geologic and geochemical field studies of radon in rocks, soils, and water. U.S. Geological Survey Bulletin, 38–49.

Hasan, M. K., Burgess, W., & Dottridge, J. (1999). The vulnerability of the Dupi Tila aquifer of Dhaka, Bangladesh. IAHS-AISH Publication, 259, 91–98.

Kareem, D. O., Ibrahim, A. A., & Ibrahiem, O. S. (2020). Heavy metal and radon gas concentration levels in Khasa River in Kirkuk City (NE Iraq) and the associated health effects. Arabian Journal of Geosciences, 13(19). https://doi.org/10.1007/s12517-020-06037-8

Khan, M. A., Khattak, N. U., Hanif, M., Al-Ansari, N., Khan, M. B., Ehsan, M., & Elbeltagi, A. (2022). Health risks associated with radon concentrations in carbonate and evaporite sequences of the uranium-rich district Karak, Pakistan. Frontiers in Environmental Science, 10, 1814. https://doi.org/10.3389/fenvs.2022.1020028

Khan, R., Islam, M. S., Tareq, A. R. M., Naher, K., Islam, A. R. M. T., Habib, M. A., Siddique, M. A. B., Islam, M. A., Das, S., Rashid, M. B., Ullah, A. K. M. A., Miah, M. M. H., Masrura, S. U., Bodrud-Doza, M., Sarker, M. R., & Badruzzaman, A. B. M. (2020). Distribution, sources and ecological risk of trace elements and polycyclic aromatic hydrocarbons in sediments from a polluted urban river in central Bangladesh. Environmental Nanotechnology, Monitoring & Management, 14, 100318. https://doi.org/10.1016/J.ENMM.2020.100318

Khatun, M., Ali, R. M. E., Karim, S., & Munsura Akther, K. (2019). Geomorphology and Geology of the Dhaka City Corporation Area-an Approach of Remote Sensing and GIS Technique. International Journal of Astronomy, 6(2), 7–16.

Majumder, R. K., Das, S. C., Rasul, M. G., Khalil, M. I., Dina, N. T., Kabir, M. Z., Deeba, F., & Rajib, M. (2021). Measurement of radon concentrations and their annual effective doses in soils and rocks of Jaintiapur and its adjacent areas, Sylhet, North-east Bangladesh. Journal of Radioanalytical and Nuclear Chemistry, 329(1), 265–277. https://doi.org/10.1007/s10967-021-07771-3

Mostečak, A., Perković, D., Kapor, F., & Veinović, Ž. (2018). Radon mapping in croatia and its relation to geology. In Rudarsko Geolosko Naftni Zbornik (Vol. 33, Issue 3, pp. 1–11). University of Zagreb, Faculty of Political Sciences. https://doi.org/10.17794/rgn.2018.3.1

Mustapha, A. O., Patel, J. P., & Rathore, I. V. S. (2002). Preliminary report on radon concentration in drinking water and indoor air in Kenya. Environmental Geochemistry and Health, 24(4), 387–396. https://doi.org/10.1023/A:1020550103471

Opondo, K. M., & Sims, K. (2012). Electronic Radon Detector User Manual. https://durridge.com/documentation/RAD7 Manual.pdf

Rajashekara, K. M., Narayana, Y., & Siddappa, K. (2007). 222Rn concentration in ground water and river water of coastal Karnataka. Radiation Measurements, 42(3), 472–478. https://doi.org/10.1016/j.radmeas.2006.12.010

Rani, S., Kansal, S., Singla, A. K., & Mehra, R. (2021). Radiological risk assessment to the public due to the presence of radon in water of Barnala district, Punjab, India. Environmental Geochemistry and Health, 43(12), 5011–5024. https://doi.org/10.1007/s10653-021-01012-y

Shivanandappa, K. C., & Yerol, N. (2018). Radon concentration in water, soil and sediment of Hemavathi River environments. Indoor and Built Environment, 27(5), 587–596. https://doi.org/10.1177/1420326X16688522

Thumvijit, T., Chanyotha, S., Sriburee, S., Hongsriti, P., Tapanya, M., Kranrod, C., & Tokonami, S. (2020). Identifying indoor radon sources in Pa Miang, Chiang Mai, Thailand. Scientific Reports, 10(1), 1–14. https://doi.org/10.1038/s41598-020-74721-6

UNSCEAR. (2010). SOURCES AND EFFECTS OF IONIZING RADIATION United Nations Scientific Committee on the Effects of Atomic Radiation: Vol. I (Issue c).

WHO. (2017). Guidelines for drinking-water quality, 4th edition.

Yong, J., Liu, Q., Wu, B., Hu, Y., & Feng, G. (2021). Assessment of radiation dose hazards caused by radon and its progenies in tap water by the human dosimetric model. Journal of Water and Health, 19(6), 933–945. https://doi.org/10.2166/wh.2021.113

---

## [Decision Letter · Decision Letter 1]

22 Mar 2023

PONE-D-22-28802R1A study on measuring the 222Rn in the Buriganga River and tap water of the megacity DhakaPLOS ONE

Dear Dr. Yeasmin,

Thank you for submitting your manuscript to PLOS ONE. After careful consideration, we feel that it has merit but does not fully meet PLOS ONE’s publication criteria as it currently stands. Therefore, we invite you to submit a revised version of the manuscript that addresses the points raised during the review process.

We look forward to receiving your revised manuscript.

Kind regards,

Sakae Kinase, Ph.D.

Academic Editor

PLOS ONE

Journal Requirements:

Reviewers' comments:

Reviewer's Responses to Questions

**Comments to the Author**

1. If the authors have adequately addressed your comments raised in a previous round of review and you feel that this manuscript is now acceptable for publication, you may indicate that here to bypass the “Comments to the Author” section, enter your conflict of interest statement in the “Confidential to Editor” section, and submit your "Accept" recommendation.

Reviewer #1: All comments have been addressed

Reviewer #3: All comments have been addressed

2. Is the manuscript technically sound, and do the data support the conclusions?

Reviewer #1: Yes

Reviewer #3: Yes

3. Has the statistical analysis been performed appropriately and rigorously? 

Reviewer #1: Yes

Reviewer #3: Yes

4. Have the authors made all data underlying the findings in their manuscript fully available?

Reviewer #1: Yes

Reviewer #3: Yes

5. Is the manuscript presented in an intelligible fashion and written in standard English?

Reviewer #1: Yes

Reviewer #3: Yes

6. Review Comments to the Author

Reviewer #1: (No Response)

Reviewer #3: The manuscript has been updated nicely. However, a few points below are still necessary as minor corrections:

1) Although a new paragraph regarding the ‘Radiological risks based on geology’ has been added; the impact on the research area of Dhaka city (also surrounding areas) seemed to be missing. It would be better if some discussion on geology and Dhaka city is given specifically in the text.

2) In the dose calculation equation, CW (daily water consumption) = 3 L/day is used. How suitable is it based on Dhaka city/Bangladeshi people’s lifestyle?

3) Scientific purposes are still not adequately discussed. The conclusion part should introduce and focus on the new findings. Recommendations given in the conclusion part are not based on the findings of this research; they are too general. In this stage, this document would be Review Paper, not Original Paper.

4) Comparison/association of tap water and river water for concentration and dose have not been sufficiently discussed from environmental/scientific viewpoints.

5) Lower detection limit has newly been clarified in the revised manuscript. However, that was based on other studies. It would be better to use the value using the detector, which was used in your research considering your measurement condition.

7. PLOS authors have the option to publish the peer review history of their article (what does this mean?). If published, this will include your full peer review and any attached files.

Reviewer #1: No

Reviewer #3: No

---

## [Author Response · Author response to Decision Letter 1]

7 May 2023

Response to academic editor and reviewer(s) 

Respected Academic Editor and Reviewers,

Thank you so much for your comments. We are very grateful to you for the time and intelligence that you have shared with us. It is a great learning opportunity for us through these reviews. Our responses to the comments are highlighted in red color in the revised manuscript.

Comments and Suggestions for Authors:

Reviewer #3 comment: The manuscript has been updated nicely. However, a few points below are still necessary as minor corrections: 

Reviewer 3, comment 1: Although a new paragraph regarding the ‘Radiological risks based on geology’ has been added; the impact on the research area of Dhaka city (also surrounding areas) seemed to be missing. It would be better if some discussion on geology and Dhaka city is given specifically in the text.

Response: Thank you very much for your suggestion. As per your suggestions, we have taken the section 2.3 ‘Radiological risks based on geology’ in the result and discussion part under the section “3.3 Radiological risks based on geology of the study area”, (page no. 15, line no. 318-353), where the possible impact of radon is discussed based on the geology of Dhaka city and its periphery region. 

Reviewer 3, comment 2: In the dose calculation equation, CW (daily water consumption) = 3 L/day is used. How suitable is it based on Dhaka city/Bangladeshi people’s lifestyle?

 Response: We thank the reviewer for this valuable comment. We have added a citation about the daily water consumption of Bangladeshi people. (page no. 08, line no. 213). 

Reviewer 3, comment 3: Scientific purposes are still not adequately discussed. The conclusion part should introduce and focus on the new findings. Recommendations given in the conclusion part are not based on the findings of this research; they are too general. In this stage, this document would be Review Paper, not Original Paper.

Response: We thank the reviewer for the opportunity to clarify this issue. We have added the following lines in the results section (page no.15-16, line no.354-369), “The Stochastic radiation model is based on the probabilistic nature of radiation-induced cancer and suggests that there is no threshold limit for radiation exposure below which the risk of cancer becomes zero. This means that even a single atom of 222Rn in water can potentially cause severe damage to the body by ionizing molecules and damaging cellular structures. Therefore, it is important to closely monitor the levels of 222Rn in water, as even low concentrations can pose a risk to human health. Despite all measured values of 222Rn levels in the tap and river water of Dhaka being below the limit set by the USEPA and WHO, continuous monitoring is essential to ensure that the levels remain within safe limits. The USEPA limit for 222Rn in drinking water is 11.1 Bq/L, while the WHO guideline value is 100 Bq/L. In this study, the measured levels of 222Rn in tap water and river water ranged from 0.56 ± 0.30 to 3.06 ± 0.61 Bq/L and from 0.35 ± 0.18 to 1.16 ± 0.61 Bq/L, respectively. The corresponding effective doses were found to be below the limit of 0.1 mSv/y recommended by the WHO [8]. Nevertheless, given the potential health risks associated with even small amounts of 222Rn in water, continuous monitoring of its levels is essential to ensure public safety. This study on 222Rn levels in Dhaka's water may provide valuable insights for future research on radiation exposure and human health.”

We have added the following lines in the recommendation section (page no. 17, line no. 400-414)

“A few recommendations are proposed for future 222Rn-related works,

• Expansion of the study to other regions of Bangladesh to get a better understanding of the national distribution of 222Rn levels in different dwelling media, including water, air, and soil is essential.

• It is necessary to investigate the potential impact of local geological factors, such as soil type and groundwater composition, on the levels of 222Rn in water sources.

• Exploration of the relationship between 222Rn exposure and cancer incidence rates in the region is crucial to gain a better understanding of the public health implications of 222Rn exposure in Dhaka and other regions of Bangladesh.

• A risk assessment study should be conducted to evaluate the potential health risks associated with chronic exposure to low levels of 222Rn in water sources in Dhaka and other regions of Bangladesh.

• A long-term monitoring program needs to be developed and implemented to track changes in 222Rn levels over time and ensure that public health safety measures remain effective.”

Reviewer 3, comment 4: Comparison/association of tap water and river water for concentration and dose have not been sufficiently discussed from environmental/scientific viewpoints.

 Response: We are grateful to the reviewer for this comment. We have already mentioned some information relevant to the environmental/scientific viewpoints in the introduction section (page no. 03, line no. 77-85). However, as per your suggestion, we have added the following sentences in the results and discussion section (page no. 16, line no. 369-389), “Many advanced countries have established national reference limits for radon in water and indoor air in order to ensure radiological safety and protect public health. However, Bangladesh currently lacks such a reference level for 222Rn in water, despite millions of people in the Dhaka megacity relying solely on tap water for daily household activities, including washing, bathing, drinking, and cooking. Given that the Buriganga river serves as a major transportation hub and facilitates to many businesses and trade centers, there is a greater likelihood of 222Rn exposure for the general population. In terms of concentration, it has been observed in this study that, tap water has a higher concentration of 222Rn than the river water. This is because, radon in river water can be easily diluted due to greater surface and interactions. However, this can vary depending on factors such as the geology of the area and the treatment processes used for tap water. When it comes to dose, the risk of exposure to 222Rn from tap water is greater than from river water, as people are likely consume more tap water than river water. However, exposure to 222Rn in river water can still occur through activities such as swimming and fishing. Overall, from an environmental and scientific viewpoint, it is important to measure and monitor the concentration of 222Rn in both tap water and river water to ensure that exposure levels do not exceed safe limits. This can help to protect public health and ensure that the water we use for daily activities is safe and free from harmful contaminants. Therefore, this study measures the 222Rn levels in both tap water and Buriganga river water to determine if they fall within the safe limits and ultimately ensure the radiological safety of the public. This study on 222Rn levels in Dhaka city's water may provide valuable insights for future research on radiation exposure and human health.”

Reviewer 3, comment 5: Lower detection limit has newly been clarified in the revised manuscript. However, that was based on other studies. It would be better to use the value using the detector, which was used in your research considering your measurement condition.

 Response: We thank the reviewer for indicating this issue. The lower detection limit of our detector is consistent with the cited paper, that is why we cited that paper.

---

## [Editor Report · Decision Letter 2]

12 May 2023

A study on measuring the 222Rn in the Buriganga River and tap water of the megacity Dhaka

PONE-D-22-28802R2

Dear Dr. Yeasmin,

We’re pleased to inform you that your manuscript has been judged scientifically suitable for publication and will be formally accepted for publication once it meets all outstanding technical requirements.

Kind regards,

Sakae Kinase, Ph.D.

Academic Editor

PLOS ONE

Additional Editor Comments (optional):

I have much pleasure in recommending this paper for publication. This manuscript has been substantially with changes highlighted point by according to reviewer's comments. The authors revised and improved the manuscript a suggested.
---

## [Editor Report · Acceptance letter]

15 May 2023

PONE-D-22-28802R2 

A study on measuring the ^222^Rn in the Buriganga River and tap water of the megacity Dhaka 

Dear Dr. Yeasmin:

I'm pleased to inform you that your manuscript has been deemed suitable for publication in PLOS ONE. Congratulations! Your manuscript is now with our production department. 

Kind regards, 

on behalf of

Professor Sakae Kinase 

Academic Editor

PLOS ONE